# In Vivo Nutritional Assessment of the Microalga *Nannochloropsis gaditana* and Evaluation of the Antioxidant and Antiproliferative Capacity of Its Functional Extracts

**DOI:** 10.3390/md20050318

**Published:** 2022-05-11

**Authors:** Rosario Martínez, Alejandro García-Beltrán, Garyfallia Kapravelou, Cristina Mesas, Laura Cabeza, Gloria Perazzoli, Palmira Guarnizo, Alberto Rodríguez-López, Roberto Andrés Vallejo, Milagros Galisteo, Pilar Aranda, Jose Prados, María López-Jurado, Consolación Melguizo, Jesus M. Porres

**Affiliations:** 1Department of Physiology, Institute of Nutrition and Food Technology (INyTA), Biomedical Research Center (CIBM), Universidad de Granada, Avda del Conocimiento s/n, 18100 Granada, Spain; rosariomz@ugr.es (R.M.); alejandrogb@ugr.es (A.G.-B.); kapravelou@ugr.es (G.K.); paranda@ugr.es (P.A.); jmporres@ugr.es (J.M.P.); 2Department of Anatomy, Institute of Biopathology and Regenerative Medicine (IBIMER), Biomedical Research Center (CIBM), Universidad de Granada, Avda del Conocimiento s/n, 18100 Granada, Spain; cristinam@ugr.es (C.M.); lautea@ugr.es (L.C.); gperazzoli@ugr.es (G.P.); jcprados@ugr.es (J.P.); melguizo@ugr.es (C.M.); 3Biosanitary Institute of Granada (IBS GRANADA), 18014 Granada, Spain; 4Microalgas Carboneras, S.L. Parque Científico-Tecnológico de Almería (PITA), C/Albert Einstein 15, 04131 Almería, Spain; palmiraguarnizo@algavillage.com; 5CIEMAT, Energy Efficiency in Building Unit, Energy Department, Renewable Energy Division, Avda. Complutense 40, 28040 Madrid, Spain; alberto.rodriguez@ciemat.es; 6Endesa Generación, S.A. Ribera del Loira 60, 28042 Madrid, Spain; roberto.andres@enel.com; 7Department of Pharmacology, School of Pharmacy, University of Granada, 18071 Granada, Spain; mgalist@ugr.es

**Keywords:** *Nannochloropsis gaditana*, functional extracts, nutrient bioavailability, antioxidant capacity

## Abstract

*Nannochloropsis gaditana* is a microalga with interesting nutritional and functional value due to its high content of protein, polyunsaturated fatty acids, and bioactive compounds. However, the hardness of its cell wall prevents accessibility to these components. This work aimed to study the effect of a treatment to increase the fragility of the cell wall on the bioavailability of its nutrients and functional compounds. The antioxidant and antiproliferative capacity of functional extracts from treated and untreated *N. gaditana* was assessed, and the profile of bioactive compounds was characterized. Furthermore, to study the effect of treatment on its nutrient availability and functional capacity, an in vivo experiment was carried out using a rat experimental model and a 20% dietary inclusion level of microalgae. Functional extracts from treated *N. gaditana* exhibited higher antioxidant activity than the untreated control. Furthermore, the treated microalga induced hypoglycemic action, higher nitrogen digestibility, and increased hepatic antioxidant activity. In conclusion, *N. gaditana* has interesting hepatoprotective, antioxidant, and anti-inflammatory potential, thus proving itself an ideal functional food candidate, especially if the microalga is treated to increase the fragility of its cell wall before consumption.

## 1. Introduction

In recent years, health-related aspects of food have acquired special importance for the consumer, and numerous functional foods from different sources have been developed. Among them, those exerting beneficial effects on obesity, cancer, and metabolic disorders such as metabolic syndrome, major epidemics in current society, have attracted special attention. Furthermore, hepatic functionality in alterations such as non-alcoholic fatty liver disease can be improved by plant-derived foods or functional extracts [1]. Such increased interest is reflected in a growing number of new patents for the food industry [2]. Research on market trends as well as the discovery of novel potential raw materials is essential for the development of new fifth range products.

In this regard, plant-based products, including microalgae, are of particular relevance due to their valuable content of protein and peptides, together with other components such as pigments, lipids, vitamins, minerals, and bioactive compounds, with possible application in different fields, mainly the food industry and the biomedical field [3,4]. They have received interest as potential alternative protein sources and are also studied as a source of safe antioxidants to prevent oxidative deterioration of food and to minimize oxidative damage to living cells. Although the occurrence of phenolic compounds in plants is well known and this group of compounds exhibits antioxidant activity in biological systems [5], the antioxidant properties of microalgae consumption are still largely unknown.

The main antioxidants found in algae are vitamins C and E [6], carotenoids (β- and α-carotene, zeaxanthin, and neoxanthin), chlorophylls [7,8], and phenols [9]. In addition, microalgae are also interesting protein sources reaching levels of up to 50% *w/w* [10]. The amino acid pattern of almost all these organisms compares favorably to that of other plant proteins (wheat and soybean) due to their contribution of essential amino acids [11]. The enzymatic hydrolysis of proteins is used to improve their functional properties without affecting their nutritional value, and this has drawn attention recently due to the generation and function of some bioactive peptides, particularly for the antioxidant activity that has been widely reported [12]. The unsaturated fatty acids are bioactive compounds that provide microalgae with high added value [13]. These components, together with metabolites such as phenols, folate, folic acid, and other pigments, can produce health benefits related to the treatment of hypercholesterolemia, cardiovascular disease, viral infections, and cancer [6]. Specifically, eicosapentaenoic (EPA) and docosahexaenoic (DHA) acids have become very popular for producing nutraceuticals containing omega-3 fatty acids.

Among the different microalgal cultures used for research or by the industry, *N. gaditana*, a non-genetically modified marine microalga isolated for the first time by Lubian [14] from the Bay of Cadiz, is of special interest. At the nutritional level, this microalga exhibits an interesting composition of protein; carbohydrates; polyunsaturated fatty acids; minerals; vitamins; and several bioactive components with potential health benefits such as chlorophyll, β-carotene, astaxanthin, canthaxanthin, violaxanthin, and zeaxanthin with high antioxidant capacity. The different products obtained can be used in the preparation of biofuel, aquaculture, animal nutrition, and human nutrition or in the generation of cosmetics, nutraceuticals, and pharmaceuticals (polysaccharides, phycobiliproteins, lipids, active proteins, glycerol, polysaccharides, and pigments).

*N. gaditana* is mainly used as live food in aquaculture due to its high content of polyunsaturated fatty acids, although it is a good source of pigments as well. Nevertheless, the aspects related to human nutrition deserve special emphasis, since, despite the nutritional potential of the microalga, there is relatively little information on the real bioavailability of its nutrients, its lack of toxicity, and its potential for the production of bioactive extracts that show beneficial effects on human health. A potential drawback to wider use of *N. gaditana* in human nutrition is its thick cell wall structure (made up of cellulose, polysaccharides, proteins, and lipids) that shows strong resistance to mechanical and chemical treatments [15,16], but new technological methods have been reported to produce viable products [17]. On the other hand, the identification of microalgal bioactive compounds is a complex task that requires multidisciplinary approaches. Currently, integrated technologies for the cultivation of microalgae are being sought to isolate various biologically active substances from biomass to increase the profitability of its production [18].

Given the background described above, this study aimed to (i) study the composition of nutrients and bioactive compounds of *Nannochloropsis gaditana* and assess the effect of a technological treatment to induce the breakdown of the cell wall on the antioxidant and antiproliferative capacity of the microalga, (ii) test the effect of the above-mentioned technological treatment on the bioavailability of protein and fat in an in vivo rat experimental model, and (iii) assess the influence of microalgae consumption on metabolic parameters of kidney and liver functionality. Considering the potential of the microalgal extracts as functional foods, we focused our study on the antioxidant and antiproliferative activities of the microalgal extracts and the influence of microalgal consumption on liver functionality and its relationship with different metabolic parameters of plasma and urine, as well as with the biotransformation of essential fatty acids and their incorporation into the liver, plasma, and erythrocytes.

## 2. Results

### 2.1. Nutritional Composition and Fatty Acid Profile of Treated and Untreated *N. gaditana*

The nutritional composition of treated and untreated *N. gaditana* is presented in Table 1. The untreated microalga was characterized by high protein (44.2%), fat (15%), and ash (19%) content. Concerning the fatty acid profile of *N. gaditana*, it was characterized by the highest levels of PUFAs (48.2%), followed by MUFAs (26.6%) and SFAs (25.2%). Specifically, eicosapentaenoic acid (EPA) exhibited the highest content, followed by palmitoleic and palmitic acid. The technological treatment of high pressure and temperature designed to make the microalga cell wall more fragile did not alter the total protein and ash composition of treated vs. raw *N. gaditana*, but considerably reduced its total fat content and markedly altered the fatty acid profile, considerably increasing the amount of SFAs and MUFAs at the expense of PUFAs.

### 2.2. Chemical Characterization and Antiproliferative Activity of the Ethanolic Extracts from Untreated and Treated *N. gaditana*

Extraction yields and antioxidant activity of ethanolic extracts from untreated or treated *N. gaditana* are presented in Table 2. In general, the technological treatment of *N. gaditana* caused a significant increase in extraction yield expressed as mg extract/mL or mg extract/g microalgae, as well as in total polyphenol content and total antioxidant activity measured by DPPH, ABTS, or lipid peroxidation inhibition, either expressed per mg of ethanolic extract or gram of microalgae. The greatest gain induced by the treatment process was attained for total polyphenol content (13.9-fold increase in µg GA/g of microalgae).

The ethanolic extracts from untreated or treated *N. gaditana* were assayed in a T84 colon cancer cell line, as well as in CCD18 colon or MCF-10A mammary non-tumor cell lines, to assess their antiproliferative activity. As shown in Figure 1, the antiproliferative activity of treated *N. gaditana* extract was significantly reduced compared to the untreated microalga as seen by the significantly higher IC50 achieved by the former in the tumoral cell line T-84 and the absence of IC50 in the CCD18 and MCF-10A non-tumor cell lines. Regarding untreated *N. gaditana* extract, a lower IC50 value was obtained in CCD18 vs. T84 colon cell line, whereas the highest IC50 was obtained for the MCF-10A cell line.

### 2.3. In Vitro Digestibility of Antioxidant and Antiproliferative Activity from Untreated or Treated *N. gaditana*

The effects of technological treatment on the potential availability of antioxidant and antiproliferative activity from *N. gaditana* after an in vitro digestion process are presented in Table 3. Both microalgae exhibited interesting potential digestibility of dry matter and antioxidant capacity expressed as total polyphenol content, ABTS, or inhibition of lipid peroxidation. Nevertheless, treated *N. gaditana* showed greater values for dry matter and the three antioxidant capacity tests in the potentially absorbable fraction (dialyzate) compared to untreated control, but only higher total polyphenol content in the less digestible fraction (retentate), whereas no significant differences were observed in the other two assays. Regarding the antiproliferative activity, it was only detected in the retentate of treated *N. gaditana*.

### 2.4. Identification and Potential Digestibility of Bioactive Compounds from Untreated or Treated *N. gaditana*

The profiles of bioactive compounds from treated or untreated *N. gaditana* identified in the ethanolic extracts of the microalgae or the dialyzate after their in vitro digestibility process are presented in Table 4. Eight bioactive compounds of different chemical nature were identified in untreated *N. gaditana* extract, whereas ten were identified in treated *N. gaditana*. Technological treatment of the microalgae resulted in the identification of new compounds compared to the untreated sample. Biological activity differed among the compounds identified from both samples, including the antiproliferative/cytotoxic action of trihydroxyanthraquinone, the marine sesterterpenoid-type metabolite heteronemin, the marine-derived macrolide compound lasonolide G, the withanolide saponins sitoindoside IX and withalongolide J, and the ergostane-type steroid physapubenolide; the antioxidant activity of polyphenolic compounds such as epicatechin 3-O-(4-methylgallate), 3-methylflavone-8-carboxylic acid, caffeic acid 3-glucoside, and quercetin-3-O-malonylglucoside; the anti-inflammatory action of the iridoid monoterpenoid asperulosidic acid and the eunicellin diterpenoid klymollin E; and the antibiotic effects of the marine bioactive polyacetylene petrosynone and the polyether monensin B. Regarding the effect of in vitro digestion, of the three compounds identified in the dialyzate of untreated *N. gaditana*, two of them, caffeic acid 3-glucoside and 3-o-(α-L-oleandrosyl) oleandolide, were not present in the microalgae while the third one, heteronemin, had been previously identified. In the case of treated *N. gaditana*, two of the three compounds identified in the dialyzate, quercetin-3-O-malonylglucoside and petrosynone, were already present in the microalgae preparation, whereas lasonolide G was not.

### 2.5. In Vivo Experiments

#### 2.5.1. Composition of the Experimental Diets

The formulation and composition of the experimental diets are presented in Table 5. Treated and untreated *N. gaditana* were included at a 20% inclusion level in the experimental diets, giving rise to considerably higher protein and total mineral content, as well as slightly higher total fat content of the former diets, compared to the casein control, whereas their carbohydrate content was markedly reduced. As a result of such changes, caloric content calculated for the experimental diets was slightly inferior to the control.

#### 2.5.2. Nutritive Utilization of Protein

The effects of dietary inclusion of treated and untreated *N. gaditana* on food intake, protein efficiency ratio, food transformation index, and digestive and metabolic utilization of protein are presented in Table 6. There were no significant differences in food or energy intake among the three diets tested. In contrast, total mineral intake (data not shown) and fecal weight were significantly higher in the animals fed microalgae-containing experimental diets compared to the control. Regarding body weight gain, no significant differences were apparent among the three diets tested, although the protein efficiency ratio and food transformation index, indices of nutritional assessment of protein and food efficiency, were significantly lower in microalgae-containing diets compared to the control and exhibited poorer values in treated vs. untreated *N. gaditana* formulations. The amount of ingested, fecal, and urinary nitrogen was significantly higher in experimental *N. gaditana* diets compared to the control, and no significant differences were apparent between the two microalgae diets in N intake and urinary excretion, whereas fecal excretion was significantly lower for treated vs. untreated microalgae. Such differences in fecal N gave rise to significantly higher digestive utilization of the former nutrient in treated vs. untreated *N. gaditana* either expressed as net absorption or apparent digestibility coefficient (ADC). Nevertheless, the digestive utilization of N in both experimental diets of microalgae expressed as the latter index was inferior to that obtained for the casein control, whereas net absorption of the nutrient was higher in microalgae diets. Regarding metabolic utilization of protein expressed as net N retention, it was lower in the untreated *N. gaditana* diet compared to the treated microalgae or control diets, among which no significant differences were found. When metabolic utilization of protein was expressed as the percentage of retained vs. absorbed N, treated *N. gaditana* diet again exhibited significantly higher values than untreated microalgae, although both experimental diets showed a significantly lower index compared to the control.

Changes in fecal morphology/microstructure of rats fed the different experimental diets are presented in Figure 2. The microstructure of feces from the animals fed untreated *N. gaditana* clearly shows the presence of undigested microalgae that disappeared as a result of the technological treatment of the microalgae. Furthermore, a significantly greater number of microbes was apparent in the treated diet when compared to the untreated *N. gaditana* or control diets.

#### 2.5.3. Fatty Acid Profile of Plasma, Erythrocyte Membrane, and Liver

The fatty acid profile of plasma, erythrocytes, and liver was characterized by the presence of saturated (SFAs: palmitic and stearic), monounsaturated (MUFAs: palmitoleic, oleic, and octadecenoic), and polyunsaturated (PUFAs: linoleic, arachidonic, and docosahexaenoic (DHA)) fatty acids (Table 7). Some exceptions could be identified, such as the presence of monounsaturated erucic acid only in the liver, the polyunsaturated eicosapentaenoic (EPA) acid only in plasma and erythrocytes, or the monounsaturated nervonic acid only present in erythrocytes and liver. No significant differences in the proportion of MUFAs or PUFAs were observed among the microalgae-containing diets vs. control, whereas only minor, although significant, increments were observed in the percentage of SFAs. Nevertheless, significant changes were observed in the profile of some essential fatty acids. Specifically, the levels of eicosapentaenoic acid were significantly increased in plasma and erythrocytes of rats fed untreated but not treated *N. gaditana*. A similar trend was obtained for erucic acid in the liver and nervonic acid in erythrocytes and liver. Other specific changes associated with the consumption of untreated *N. gaditana* were the significant decrease in plasma levels of arachidonic acid or liver levels of oleic acid. Concerning docosahexaenoic acid, a trend towards a general increase in its levels resulting from microalgae consumption was observed in plasma and liver, although significant differences were only observed in the latter. Likewise, the percentage of hepatic stearic acid was significantly increased by the consumption of treated or untreated *N. gaditana*.

#### 2.5.4. Glucose Metabolism and Organ Weight

The effects of dietary inclusion of untreated or treated *N. gaditana* on glycemic profile and area under the curve (AUC) after an oral glucose overload carried out at the end of the experimental stage are presented in Figure 3. Daily ingestion of untreated *N. gaditana* resulted in significantly higher AUC, higher postprandial blood glucose peak after 15 min of administration, and higher values of glycemia during a more prolonged period after glucose overload when compared to the control or treated *N. gaditana* groups.

Consumption of treated or untreated *N. gaditana* caused varying effects on the weight of different organs and tissues (Table 8). Of special relevance was their significantly hypertrophic action on the cecum and kidney, as well as the decrease in weight of epididymal and abdominal fat, although significant differences were only apparent in the latter tissue.

#### 2.5.5. Hematic and Plasma Biochemical Parameters

The influence of dietary inclusion of *N. gaditana* on hematic and biochemical parameters of rats is presented in Table 9. Hemoglobin content and hematocrit were significantly higher in rats fed untreated *N. gaditana* compared to the control, but not in rats fed the treated microalgae. MCV was significantly higher in both microalgae diets compared to the control. Plasma HDL-C increased as a result of microalgae consumption but results were only significantly higher after consumption of treated *N. gaditana*. Plasma ALT and ALP activities were also increased by microalgae consumption, but differences vs. control diet were only significant for the group of rats fed untreated *N. gaditana*. Plasma P was significantly decreased in animals fed microalgae compared to the controls.

#### 2.5.6. Urinary Parameters of Kidney Functionality

Changes in urinary parameters of metabolic and renal functionality as a result of the different dietary treatments are shown in Table 10. Urinary pH was significantly lower for animals fed treated or untreated microalgae compared to the control. In contrast, urinary volume was significantly higher in the experimental groups compared to the control, and the highest excretion was found for animals fed untreated *N. gaditana*. Total P excretion was also higher in the experimental vs. control group, but the highest mineral excretion was found in animals fed the treated microalgae. Regarding uric acid and albumin excretion, ingestion of microalga led to a significant decrease in their urinary content compared to the control. On the other hand, creatinine excretion was lowest in the group of rats fed untreated *N. gaditana*, whereas no significant differences were apparent between CT and NGT groups.

#### 2.5.7. Liver Fat Content and Antioxidant Capacity

Consumption of untreated or treated *N. gaditana* caused a slight increase in liver weight (2.69 g/100 g body weight in the control group vs. 3.19 and 3.39/100 g body weight in animals fed untreated or treated *N. gaditana*). In contrast, the hepatic fat content was significantly decreased by the dietary treatment with microalgae (Figure 4), while no significant differences were observed in the former parameter between the dietary inclusion of treated vs. untreated *N. gaditana*. Regarding hepatic antioxidant capacity, consumption of *N. gaditana* induced an increment in the enzymes assayed, although the effect was more pronounced in the treated microalgae that significantly increased the activity of the four enzymes tested when compared to the untreated microalgae that only caused a significant increase in Cu/Zn-SOD and catalase activity vs. control diet while it decreased the activity of GPX (Table 11).

## 3. Discussion

The results of the present study represent a new approach to the potential benefits of *N. gaditana* consumption either as a nutritional supplement or in the prevention and treatment of diseases such as cancer, metabolic syndrome, and non-alcoholic fatty liver disease. The study sheds new light on the concept of this microalga as a functional ingredient since, in addition to its nutritional value, it exhibits interesting antioxidant properties as well as reduced cytotoxicity.

Among the key morphological aspects of *N. gaditana* is the rigidity of its cell wall [19]. Therefore, to improve the availability of its nutrients and bioactive components, technological treatments are sought to disrupt it. Here, a combination of temperature and pressure was implemented, and the associated changes in digestibility of certain nutrients and non-nutritional bioactive compounds were assessed, with special attention to the new functional components that appeared after the technological treatment.

The nutrient composition of *N. gaditana* used in the present study was within the range of values described in the literature [20]. Processing did not result in modifications of total protein and minerals but led to a decreased content of fat and marked changes in fatty acid profile towards a more saturated pattern. The combination of thermal/pressure processing conditions has been shown to cause an improvement in the digestibility of plant-based proteins [21]. Under our experimental conditions, it induced a positive effect on the functional value of the microalgae that was reflected in improved extraction yield, total polyphenol content, and antioxidant activity. Such effects may be more relevant if we consider that after the in vitro digestion process, treated *N. gaditana* showed greater values for dry matter and the three antioxidant capacity tests in the potentially absorbable fraction.

To identify the specific components potentially responsible for the functional activity of the untreated or treated microalgae, a phytochemical screening of bioactive compounds was carried out. It showed eight components with different chemical nature in untreated *N. gaditana* extract, some of them with known antioxidant action that were able to survive the in vitro digestion process and could be potentially accessible for intestinal absorption (caffeic acid 3-glucoside, 3-o-(α-L-oleandrosyl) oleandolide, heteroneminin). Treatment of the microalga resulted in the identification of new compounds compared to the untreated sample, and digestion of treated samples offered a different profile of potentially available bioactive compounds, some of which appeared to remain untouched by the digestion process (quercetin-3-O-malonylglucoside, petrosynone), whereas some others were most likely modifications of compounds present in the sample before the gastrointestinal digestion (lasonolide G). These bioactive compounds may be related to at least three beneficial biological properties of the microalga such as its antioxidant, antidiabetic, and antiproliferative action. In fact, the antiproliferative action of heteroneminin and the antioxidant activities of caffeic acid and quercetin-3-O-malonylglucoside have been reported [22,23,24].

The antioxidant activity of untreated or treated *N. gaditana*, assessed by measuring their total polyphenol content, DPPH, or lipid peroxidation inhibition, is most likely related to the polyphenol and carotenoid content [25,26]. Antioxidants play a crucial role in protecting cells from premature aging and disease by shielding DNA, proteins, and lipids from oxidative damage [27]. The ability of microalgae to produce antioxidant compounds has been related to their ability to survive under extremely oxidizing conditions, which increase cellular antioxidant content or trigger the production of antioxidant secondary metabolites, such as phenolic constituents [28]. The antioxidant activity of phenolic compounds is highly dependent on their chemical structure, characterized by an aromatic ring bearing one or more hydroxyl substituents [29]. The hydroxyl groups contribute to the antioxidant activity through their metal-chelating capability, as well as electron/hydrogen donation capacity, generating radical intermediates of greater chemical stability than the initial radicals. Moreover, the combined action of hydrophobic benzenoid rings and hydroxyl groups makes phenolics capable of inhibiting different enzymes involved in ROS generation, such as lipoxygenases, cyclooxygenase, and xanthine oxidase [30]. Recently, Del Mondo et al. [31] investigated the structural variety and the beneficial activity of polyphenols from microalgae, although they also highlighted the lack of genetic and biochemical information on their biosynthetic route. More studies are needed to better approach these aspects.

According to Avila-Roman et al. [32], the potential antitumor action exhibited by *N. gaditana* may be ascribed to oxylipins isolated from the microalgae (15-HEPE) that showed antiproliferative activity against UACC-62 and HT-29 cells. The oxylipins reduced ATP levels of both cell lines, suggesting a possible link with cytotoxicity. Finally, 13-HOTE was combined with the anticancer drug 5-fluorouracil, inducing a synergistic activity on HT-29 cells. Furthermore, Carrasco-Peinado et al. [33] have reported the first complete proteome analysis of *N. gaditana*, which was analyzed using the applied proteomics concept with the identification of UCA01 protein from the prohibitin family. The recombinant version of this protein showed antiproliferative activity against the tumor cell lines Caco2 and HepG-2 but did not show any effect on control cells. The low antiproliferative action found in the extracts assayed in the present experiment can be explained by the type of extraction process used to obtain the ethanolic extracts. Such low cytotoxic action of the extract was also observed in the two non-tumoral cell lines assayed in which IC50 values were also high for the untreated microalga and not available for the treated one.

A nutritional assessment study was carried out using a 20% dietary inclusion level of treated or untreated microalgae. The inclusion of *N. gaditana* led to higher protein and total mineral intake as well as higher fecal weight. Regarding the digestive utilization of protein, the significantly higher net N absorption in the microalgae-supplemented diets could be ascribed to the higher amount of protein provided by the diet. Moreover, the significant increase in N digestibility after treatment was associated with cell wall disruption and was also related to lower fecal weight as well as lower amounts of microalgae cells in feces. Treatment also improved the metabolic utilization of protein expressed either as balance or percentage of retained vs. absorbed N. Fecal microstructure deserves special attention since fecal samples from the animals fed untreated *N. gaditana* clearly show the presence of undigested microalgae that disappeared as a result of the technological treatment. Such changes were complemented with a different qualitative/visual profile of microbiota, suggesting a different pattern of cecal and colonic fermentation. Therefore, the rupture of the cell wall caused by treatment allowed the release of compounds that may have favored the growth of specific microbiota. This change, together with the increased activity of antioxidant compounds in the potentially non-absorbable fraction of microalgae already described, may contribute significantly to the protection of the colon against agents that could damage its integrity and/or functionality [34,35,36]. Dietary administration of the microalgae in the diets also led to hypertrophy of the cecum, which can be a result of the fermentation process of non-digestible components which pass through the digestive tract remaining unabsorbed [37]. These beneficial aspects at the digestive and metabolic level point to treated *N. gaditana* as a valuable protein supplement with an associated prebiotic action. Overall, these results highlight not only the important nutritive value of the microalga but also a functional effect at a digestive level.

Because of the promising results on the absorption and metabolic utilization of protein from *N. gaditana* diets and the positive impact of algae treatment, we undertook the study of fat bioavailability. Total fat from untreated *N. gaditana* was efficiently absorbed and incorporated into plasma and the erythrocyte membrane, as shown by the significantly higher levels of EPA in these two compartments exhibited by the animals fed untreated algae in which EPA is highly present. Besides, both microalgae significantly enhanced the contents of DHA in the liver. This fatty acid has been credited with interesting metabolic benefits of its derivatives in the inflammatory process, cardiovascular disease, or diabetes [38]. In contrast, the technological treatment caused a reduction in fat content and changes in the fatty acid profile towards a more saturated pattern in the treated microalgae, leading to significantly lower levels of EPA in the tissues studied compared to the untreated algae. Therefore, if the beneficial effects of *N. gaditana* n-3 fatty acids are sought, the untreated microalgae (or, alternatively, treated using a milder technological process) must be used.

Regarding the hypoglycemic action of treated vs. untreated *N. gaditana*, it can be related to the different profile of bioactive compounds from the former as well as differences in the potential availability of such compounds. Some polyphenols such as quercetin, which is potentially available from treated, but not untreated, *N. gaditana*, are well known for their ability to decrease blood glucose via different metabolic pathways [39,40]. In addition, the treated microalgae could act through the inhibition of the amylase enzyme involved in the digestion of carbohydrates [41]. *N. gaditana* per se did not result in an amelioration of glucose profile, measured by AUC, but when the microalga was treated, this effect disappeared, and the results were similar to the control group. Under our experimental conditions, the benefits of treated vs. untreated microalgae consumption highlight the usefulness of the treatment, in this case preventing an effect of the algae on glucose metabolism that could be considered non-beneficial.

The administration of the microalgae in the diets also led to hypertrophy of the kidney. The most prominent effect at the renal level is the strong increase in diuresis in the untreated group derived from the sodium content of marine microalgae; this effect was partly corrected by the treatment. There was also a strong increase in P excretion, which doubled with treatment. It is well known that the phosphorus content of microalgae is high, so a concomitant increased phosphaturia is expected. The reduction in albumin excretion in both groups supplemented with the algae points to improved renal functionality.

Regarding hepatic morphology and functionality, a small increase in liver weight was observed in rats fed with a diet supplemented with the microalgae that could be attributed to an activation of processes of xenobiotic biotransformation. Microalgae are known for their content of metals and other compounds that need to be metabolized by the liver. Liver hypertrophy was associated with an increase in ALT transaminase and ALP, which was partly corrected by treatment. On the other hand, the increase in water retention could have contributed to the increase in liver weight, as the total hepatic fat content was significantly decreased. Hepatic changes were associated with an improved plasma lipid profile with a significant increase in HDL and non-significant changes in LDL cholesterol. Likewise, hepatic functionality was beneficially affected by the intake of the diet with treated *N. gaditana*, as seen by the increased hepatic antioxidant activity, especially of Mn-SOD, Cu-Zn-SOD, and GPX. Such increased antioxidant activity in rats fed treated vs. untreated microalga was correlated with the higher values of certain parameters and indices such as the hepatic weight, net N absorption or dialyzability of dry matter, total polyphenol content, and lipid peroxidation inhibition capacity. Specifically, dialyzability of total polyphenols had the highest correlation with Mn-SOD, Cu/Zn-SOD, and GPX activity (R = 0.52, 0.96, and 0.44, respectively), whereas liver weight had the highest correlation with catalase activity (R = 0.45).

The results of this study can have a significant impact on human nutrition since novel foods are continuously being tested to increase the quality of current dietary patterns. This can be done by providing essential nutrients to improve the nutritional status of the population, or through the supplementation of bioactive compounds with beneficial health-related effects, both aspects covered by *N. gaditana*. The microalga is already commercialized although not widely distributed, and our results can give this food product added value and new potential uses.

## 4. Materials and Methods

### 4.1. Microalgal Biomass and Technological Treatment

*Nannochloroposis gaditana* was kindly provided by Endesa Generación, Carboneras, Almería, Spain. A combined thermal and high-pressure treatment, via steam explosion at temperatures between 160 and 190 °C in time intervals between 5 and 30 min, was implemented at the Instituto de la Grasa (Spanish National Research Council, Sevilla, Spain) to increase the fragility of *N. gaditana* (patent pending) and make it more susceptible to disruption during extraction processes and digestion by the experimental animals.

### 4.2. Proximal Composition of Untreated or Treated *N. gaditana*

The moisture content of microalgae flours and experimental diets was determined by drying to constant weight in an oven at 105 ± 1 °C. Total N was determined according to Kjeldahl’s method. Crude protein was calculated as N × 6.25. Total fat content was determined by gravimetry of the ether extract after acid hydrolysis of the sample. Ash content was measured by calcination at 500 °C to a constant weight. Fatty acid profile analysis was performed in aliquots of untreated or treated microalgae and their experimental diets (250 mg) that were methylated according to the method of Lepage and Roy [42] for gas chromatography analysis using an Agilent 7890A chromatograph equipped with a CTC Pal combi-xt model sampler and a Waters Quattro micro-GC mass spectrometer detector as previously described [43]. FAMEs were identified using analytical standards and a mass spectral library. Peak areas were measured and used to calculate the percentage of each fatty acid (Food Composition Analysis and Structural Assessment Analysis Units, CIC, University of Granada, Granada, Spain).

### 4.3. Ethanolic Extracts

Ethanolic extractions were carried out to obtain bioactive compounds from untreated or treated *N. gaditana*. Briefly, five grams of microalgae flour was mixed with 15 mL of hydroalcoholic extraction solution (ethanol:type I water:12N HCl; 50:50:0.25) at pH 2, 4 °C, in a nitrogen atmosphere for 30 min. Following the extraction process, the extracts were centrifuged (3000 rpm, 5 min). The supernatants were stored and the extraction process was repeated for the pellet two additional times. Finally, all the supernatants were mixed and stored at −20 °C. To evaluate the ethanolic extract yield, one-milliliter aliquots were evaporated using a Savant DNA120 vacuum evaporator (ThermoSci, Waltham, MA, USA). The evaporated extracts were freeze-dried for 24 h (Cryodos-50 lyophilizer, TELSTAR, Madrid, Spain). The dry weight of the extract was then calculated and referenced to a volume of 1 mL of initial extract, as well as to the grams of microalgae used for the extraction.

### 4.4. In Vitro Digestion of Untreated and Treated *N. gaditana*

The in vitro digestibility of untreated and treated *N. gaditana* was assessed by the methodology described by Porres et al. [44] with minor modifications [45] using a pepsin digestion period of 2 h followed by pH equilibration and pancreatin digestion for another 2 h coupled with equilibrium dialysis (MWCO 12,000–14,000 Da, Medicell International Ltd., London, UK). Briefly, 20 mL of 0.01 N HCl was mixed with 1 g of each sample, and 1 N HCl was further added to the mixture until pH 2 was reached. Then, 1 mL of pepsin solution (0.16 g/mL in 0.1 N HCl) was added for gastric digestion, and the mixture was incubated in a shaking water bath at 37 °C for 2 h. Negative controls were made with the same volume of 0.01 N HCl instead of the sample. Before the intestinal digestion, a pH compensation step of 30 min was performed with 0.1 N NaHCO_3_ added into dialysis bags which were placed in the digestion vessels in a shaking water bath at 37 °C. Then, 5 mL of a 0.1 N NaHCO_3_ solution with pancreatin (4 mg/mL) and bile salts (25 mg/mL) was added, and the mixture was incubated in a shaking water bath at 37 °C for 2 h. Once digestion was finished, the contents inside (sample dialyzed and potentially absorbable) and outside (sample retained that could potentially reach the colon) the dialysis bags were collected and kept at −20 °C to be used for the analysis of dry matter, antioxidant capacity (total polyphenol contents, inhibition of lipid peroxidation, ABTS), and antiproliferative activity.

### 4.5. Antioxidant Activity Assays

Total polyphenol content was assessed using a modified Folin–Ciocalteu colorimetric assay [46]. A 125 µL aliquot of ethanolic extracts, the in vitro digestion products, or a standard solution of gallic acid (0–600 mg/L) was mixed with 500 µL of double-distilled water and 125 µL of Folin–Ciocalteu reagent. After 6 min of incubation, 1.25 mL of a 10% (*w*/*v*) Na_2_CO_3_/1 M NaOH solution was added, and the volume was made up with water to 3 mL and incubated for 90 min. Then, the mixture was centrifuged (3000 rpm/2 min/RT), and the optical density of the supernatant was measured at 750 nm (Multiskan FC Microplate Photometer, Thermo Fisher Scientific, Waltham, MA, USA). The results were expressed as µg of gallic acid equivalent (GAE) per mg of sample.

A free radical uptake assay was performed based on the method of Miller et al. [47], who used 2,20-azino-bis(3-ethylbenzothiazoline-6-sulfonic acid) (ABTS) to measure the total antioxidant capacity of a fluid. Six microliters of ethanolic extract (with ethanol evaporated), the in vitro digestion products, or a standard solution of gallic acid (0–60 mg/L), was mixed with 294 µL of ABTS and incubated for 3 min. The optical density of the samples was then measured at 620 nm (Multiskan FC, Microplate Photometer, Thermo Fisher Scientific). The blank was made with 6 µL of water and 294 µL of ABTS. The results were expressed as µg of gallic acid equivalent (GAE) per mg of sample.

The ability of different extracts or in vitro digestion products to inhibit lipid peroxidation was assessed using thiobarbituric acid reactive substances (TBARS) measured in rat brain homogenate as described by Ohkawa et al. [48]. Brain tissue was mixed (1:10 *w*/*v*) with cold 1.15% KCl/0.1% Triton X-100 with the use of a homogenizer. The mixture was centrifuged at 2000 rpm, 4 °C, for 5 min, and the supernatant was aliquoted and stored at −20 °C for lipid peroxidation assays. Brain homogenates were treated with a mixture of FeCl_3_/H_2_O_2_ to induce lipid peroxidation, and the percentage of inhibition of TBARS formation caused by the different extracts or in vitro digestion products was determined as previously described by Kapravelou et al. [46].

A DPPH radical scavenging assay was performed according to the method of Galisteo et al. [49]. This assay determines whether ethanolic extracts can scavenge or retain free radicals generated during oxidative stress. The decrease in absorbance at 493 nm of DPPH is proportional to the activity of the antioxidant components of the samples. Briefly, to 1450 µL of a methanol solution containing 0.02 mM DPPH, 50 µL of ethanol extract or a standard solution of Trolox (0–0.5 mM) was added and left to react for 15 min. The optical density of the samples was then measured at 515 nm (Thermo Fisher Scientific). The blank was made with methanol. The results were expressed as µmol of Trolox equivalent per mg of sample.

### 4.6. Chromatographic Analysis

Ethanolic extracts of untreated or treated *N. gaditana* were analyzed by ultra performance liquid chromatography (UPLC) (Acquity H Class, Waters, Milford, MA, USA) coupled with quadrupole time-of-flight mass spectrometry (Synap G2, Waters). Ten-microliter aliquots of the extracts were injected into the chromatograph. Separation was carried out on an Acquity HSST33 analytical column (100 mm × 2.1 mm internal diameter, 1.8 μm). A gradient program that combined deionized water with 0.5% of acetic acid as solvent A and acetonitrile with 0.5% of acetic acid as solvent B was used as the mobile phase. The initial conditions were 95% A and 5% B. A linear gradient was then established to reach 95% (*v*/*v*) of B at 18 min (total run time 18 min, post-delay time 5 min, mobile phase flow rate 0.4 mL/min). Afterward, a high-resolution mass spectrometry analysis was carried out in negative electrospray ionization (ESI-ve). Spectra were recorded over the mass/charge (*m*/*z*) range of 50–1500. The retention times (RTs) and mass (MS) fragments were used to identify all the compounds. Chromatogram analysis was performed using MassLynx V4.1 software (Waters Corporation, Milford, MA, USA). The Chemspider database was used to validate the identified compounds by matching at least three of their subfragments.

### 4.7. Cell Viability Assays

Cell viability assays were performed according to the methodology described by Mesas et al. [39]. Briefly, T84 human colon adenocarcinoma and CCD18 human non-tumor colon cell lines were grown in Dulbecco’s Modified Eagle’s Medium (DMEM) (Sigma-Aldrich, Madrid, Spain) supplemented with 10% heat-inactivated fetal bovine serum (FBS) (Gibco, Madrid, Spain) and antibiotics (gentamicin/amphotericin-B+penicillin/streptomycin) (Sigma Aldrich, Madrid, Spain) at 1%. MCF-10A cell line was maintained in DMEM/F12 supplemented with 5% heat-inactivated horse serum, 0.5 μg/mL hydrocortisone, 10 μg/mL insulin, 20 ng/mL epidermal growth factor, and 100 ng/mL cholera toxin. The cell lines were maintained in an incubator at 37 °C and 5% CO_2_ humidified atmosphere. For experiments, cells were seeded in 48-well plates with DMEM (300 μL) at a density of 5 × 10^3^ cells/well. After 24 h, cell cultures were exposed to the ethanolic extract, which was previously evaporated to avoid ethanol toxicity. The extract was easily dissolved in DMEM without any additional solvent, and no signs of contamination were observed during the experiment due to the sterile nature of an ethanolic extract and the cell growth in the culture medium treated with antibiotics. Then, the cell cultures were exposed to increasing concentrations of the evaporated ethanolic extract for 72 h. After the incubation time, cells were fixed with 10% trichloroacetic acid (TCA) for 20 min at 4 °C. Once dried, the plates were stained with 0.4% sulforhodamine B (SRB) in 1% acetic acid (20 min, in agitation). After three washes with 1% acetic acid, SRB was solubilized with Trizma (10 mM, pH 10.5). Finally, the optical density (OD) at 492 nm was measured in an EX-Thermo Multiskan spectrophotometer, and the half-maximal inhibitory concentration (IC50) was calculated (GraphPad Prism 6 Software, La Jolla, CA, USA).

### 4.8. Animals and Experimental Diets

A total of 24 male Wistar rats (Charles Rives, Barcelona) aged 6 weeks with an initial average body weight of 212.8 ± 4.5 g were randomly divided into three experimental groups of eight animals each with the following design: a control group fed an AIN-93G diet with olive oil as the only fat source (CT) and two experimental groups that consumed untreated or treated *N. gaditana* at a 20% dietary inclusion level (NGUT and NGT, respectively). All diets were formulated following the recommendations to meet the nutrient requirements of the growing rat [50]. The composition of different experimental diets is presented in Table 5. The animals were housed in a well-ventilated and thermostatically controlled room (21 ± 2 °C) (Animal Experimental Unit, CIC, University of Granada), consumed the diet *ad libitum*, and had free access to type 2 water. The experiments lasted for 20 days; an initial period of 3 days was implemented to allow the animals to adapt to the experimental diet and housing conditions, followed by an experimental period of 17 days in which the animals were allocated in group cages (*n* = 4) where they were kept until the end of the experiment. Food intake was recorded daily whereas body weight was determined weekly. On week two of the experiment, eight animals from each experimental group were housed individually in metabolic cages to allow a separate collection of feces and urine for 5 days to determine the digestive and metabolic utilization of protein and fat. All experiments were undertaken according to Directional Guides Related to Animal Housing and Care [51], and all procedures were approved by the Animal Experimentation Ethics Committee of the University of Granada, Spain (Project number 16/07/2019/132). At the end of the experimental period, the animals were anesthetized with ketamine (75 mg/kg body weight) and xylazine (10 mg/kg body weight), and the blood was collected by abdominal aorta puncture using heparin as an anticoagulant. An aliquot of 0.25 mL was used to assess blood parameters (KX-21 Automated Hematology Analyzer, Sysmex Corporation), and the rest was centrifuged at 1458× *g* for 15 min to separate plasma that was subsequently frozen in liquid nitrogen and stored at −80 °C until its analysis. The erythrocyte fraction was collected, washed twice with physiological saline solution, and also stored at −80 °C. Kidneys, liver, and heart were removed and weighed to check the lack of toxicity of the microalgae on these organs. A portion of the liver was freeze-dried and processed to determine the amount of total fat and fatty acid profile.

### 4.9. Biological Indices

The following indices and parameters were determined for each group according to the formulas given below and following the methodology described by Kapravelou et al. [52]: protein efficiency ratio (PER, weight gain in grams per day/protein intake in grams per day); food transformation index (FTI, total intake in grams of dry matter per day/increase in body weight in grams per rat per day); apparent digestibility coefficient of protein (ADC) (1); nitrogen retention (nitrogen balance) (2); and percent nitrogen retention/nitrogen absorption (% R/A) (3).
ADC = [(I − F)/I] × 100(1)
balance = I − (F + U)(2)
% R/A = {[I − (F + U)]/(I − F)} × 100(3)
where I = intake, F = fecal excretion, and U = urinary excretion.

### 4.10. Blood and Plasma Biochemical Parameters

Fasting blood glucose and glucose tolerance tests performed after oral glucose overload followed the protocol described by Prieto et al. [53]. Blood glucose concentration from the animal’s tail was recorded at periods 0, 15, 30, 60, 90, and 120 min after the glucose overload ingestion (Accu-Chek Aviva, Roche), and the area under the curve (AUC) was calculated following the trapezoidal rule [54].

Plasma biochemical parameters were analyzed using a Shenzhen Midray BS-200 Chemistry Analyzer (Bio-Medical Electronics) at the Bioanalysis Unit of the Scientific Instrumentation Centre (CIC, Biomedical Research Park, University of Granada) to assess plasma lipid profile (triglycerides (mg/dL), total cholesterol, HDL-cholesterol, LDL-cholesterol (mg/dL)), albumin (g/dL), creatinine (mg/dL), aspartate aminotransferase (U/L), alanine aminotransferase (U/L), gamma-glutamyl transferase (U/L), alkaline phosphatase (U/L), and phosphorus (mg/dL).

### 4.11. Fatty Acid Profile of Plasma, Liver, and Erythrocyte Membrane

One milliliter of erythrocytes was extracted sequentially using 0.05% butyl hydroxytoluene in isopropanol, chloroform, and hexane. After hexane extraction, samples were centrifuged at 1458× *g* for 10 min. The upper phase was collected, evaporated under a nitrogen stream, and subsequently methylated. Plasma aliquots (100 µL) were directly methylated. A portion of the liver was freeze-dried, and 0.1 g was extracted and then methylated. Methylation and chromatographic analysis were carried out following the same methodology used for the determination of fatty acid profiles in microalgae and experimental diets (Section 4.2). Different product to precursor fatty acid ratios were used as indices of desaturase or desaturase–elongase enzyme activities in the liver as reported by González-Torres et al. [55] using the following formulas: Delta-6-elongase-desaturase: docosahexaenoic acid/linolenic acid (DHA/ALA), arachidonic acid/linoleic acid (ARA/LA). Stearoyl-coA activity (SCD): palmitoleic acid/palmitic acid (PE/PI), oleic acid/stearic acid (OLE/STE). Delta-5 desaturase activity: arachidonic acid/eicosatrienoic acid (ARA/EI).

### 4.12. Total Hepatic Lipid Content

A liver aliquot was freeze-dried to assess the percentage of hepatic water content. Hepatic lipids were extracted from the freeze-dried liver portion with hexane, following the methodology described by Folch et al. [56] with slight modifications [46]. Total liver lipids were measured gravimetrically after solvent evaporation under a N_2_ stream.

### 4.13. Hepatic Antioxidant Activity Assays

A fresh liver aliquot was homogenized (1:10 *w*/*v*) in 50 mM phosphate buffer (pH 7.8) containing 0.1% Triton X-100 and 1.34 mM of DETAPAC using a Micra D-1 homogenizer (ART moderne labortechnik) at 18.000 rpm for 30 s followed by treatment with Sonoplus HD 2070 ultrasonic homogenizer (Bandelin) at 50% power three times for 10 s. Liver homogenates were centrifuged at 13,000× *g*, 4 °C, for 45 min, and the supernatant was used to determine the activity of antioxidant enzymes. Catalase activity was measured using the method of Cohen et al. [57] and expressed as enzyme units calculated by the following formula: ln(A1/A2)/t, where ln is the natural log, A1 and A2 are the observed absorbances at the two selected time points, and t is the reaction time between the two points. Total cellular glutathione peroxidase (GPX) activity was determined by the coupled assay of NADPH oxidation [58] using cumene hydroperoxide as substrate. The enzyme unit was defined as nmol of NADPH oxidized per min. Total superoxide dismutase (SOD) activity was measured as described by Ukeda et al. [59]. Mn-SOD activity was determined using the same method after treating the samples with 4 mM KCN for 30 min. CuZn-SOD activity resulted from subtracting the Mn-SOD activity from the total SOD activity. One unit of SOD activity was defined as the enzyme needed to inhibit 50% 2,3-bis-(2-methoxy-4-nitro-5-sulfophenyl)-2H-tetrazolium-5-carboxanilide (XTT) reduction. Protein concentration was assayed using the Bradford assay [60].

### 4.14. Statistical Analysis

The influence of microalgae treatment on food intake; body weight; digestive and metabolic utilization of protein and fat; hematological and plasma biochemical parameters; the fatty acid profile of plasma, liver, and erythrocyte membrane; and fatty acid indices was analyzed by a one-way ANOVA. Results are given as mean values and standard error of the mean. Tukey’s test was used to detect differences between treatment means. The analyses were performed with Statistical Package for Social Sciences (IBM SPSS for Windows, version 22.0, Armonk, NY, USA), and the level of significance was set at *p* < 0.05.

## 5. Conclusions

The composition of *N. gaditana* includes substances of high nutritional value, such as polyunsaturated fatty acids, proteins, and minerals as well as a variety of functional components that can be used in the diet of humans and animals, in proportions that make it a promising novel nutritional supplement or functional food. The interesting properties of *N. gaditana* can be modified and, in many cases, enhanced by the technological treatment of pressure and heat. Potential uses of the microalga include its multi-faceted hepatoprotective activities with special emphasis on its hypolipidemic, antioxidant, and anti-inflammatory action, thus making it an ideal candidate functional food for preventing and/or managing alterations such as metabolic syndrome and associated non-alcoholic fatty liver disease.

## Figures and Tables

**Figure 1 marinedrugs-20-00318-f001:**
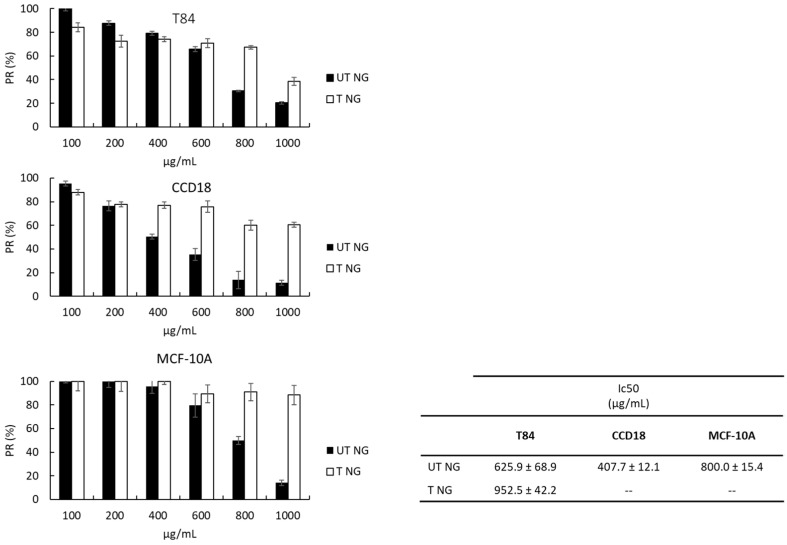
Cytotoxicity activity (IC50) of the ethanolic extracts from untreated or treated *N. gaditana* in T84, CCD18, and MCF-10A cell lines. Results are represented as the mean of three replicates ± SEM. Relative proliferation is expressed as %RP. UT NG, untreated *N. gaditana*; T NG, treated *N. gaditana*.

**Figure 2 marinedrugs-20-00318-f002:**
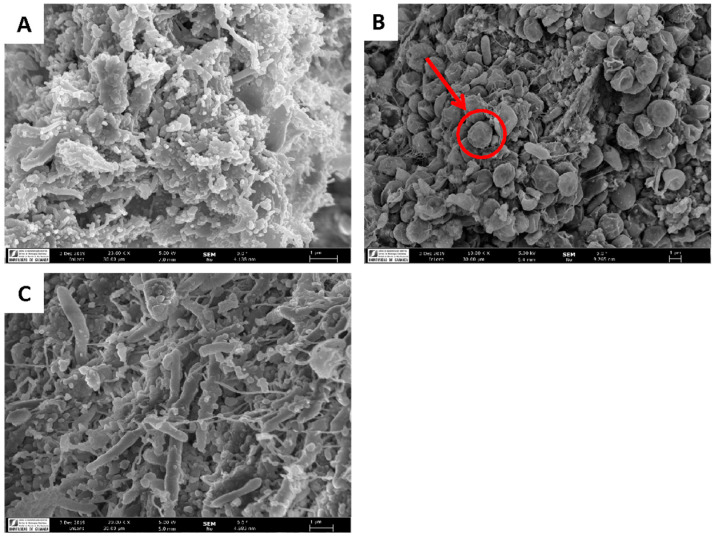
Technological treatment of *N. gaditana* affects the fecal microstructure of rats fed treated microalgae. (**A**) Electron micrograph of feces from animals fed the control diet. (**B**) Electron micrograph of feces from animals fed untreated *N. gaditana*. (**C**) Electron micrograph of feces from animals fed treated *N. gaditana*. Microalgae present in feces of animals fed untreated *N. gaditana* marked by an arrow.

**Figure 3 marinedrugs-20-00318-f003:**
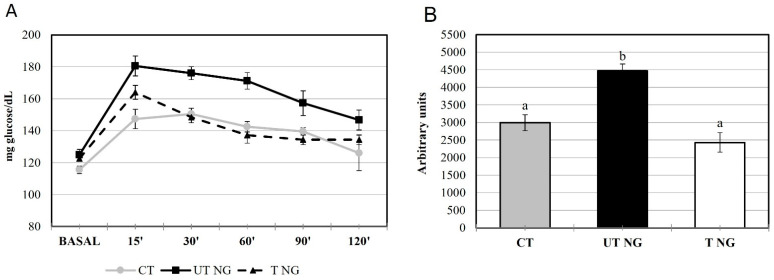
Effects of dietary treatment with *N. gaditana* on glucose metabolism. (**A**) Blood glucose values after performing an oral glucose tolerance test, (**B**) Representation of the area under the curve obtained after performing an oral glucose tolerance test. Results are the means and standard error of the mean (error bars) of 8 rats. CT, control group; UT NG, untreated *N. gaditana* group; T NG, treated *N. gaditana* group. Means within the same line with different letters differ significantly (*p* < 0.05).

**Figure 4 marinedrugs-20-00318-f004:**
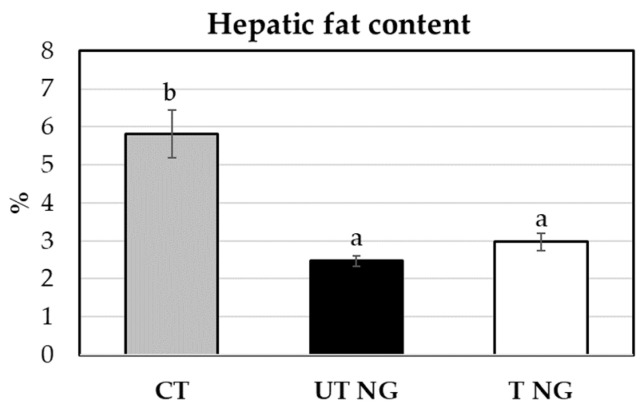
Effects of dietary treatment with *N. gaditana* on hepatic fat content. Results are the means and standard error of the mean (error bars) of 8 rats. CT, control group; NG UT, untreated *N. gaditana* group; NG T, treated *N. gaditana* group. Means within the same line with different letters differ significantly (*p* < 0.05).

**Table 1 marinedrugs-20-00318-t001:** Nutritional composition and fatty acid profile of treated and untreated *N. gaditana*.

	NG UT	NG T
** Nutritional Composition **		
Moisture	2.10	3.78
Ash	19.2	18.9
Protein	44.2	43.1
Fat	14.9	8.4
Carbohydrate	19.4	25.8
** Fatty Acid Profile (%) **		
Myristic (C14)	4.9	6.2
Palmitic (C16)	20.3	42.1
Palmitoleic (C16:1n9)	24.4	46.2
Oleic (C18:1n9)	2.2	3.5
Linoleic (C18:2n6)	2.6	0.62
Arachidonic (C20:4n6) AA (W6)	3.0	0.0
Eicosapentaenoic (C20:5n3) EPA	42.6	1.26
SFAs	25.2	48.3
MUFAs	26.6	49.8
PUFAs	48.2	1.89

NG UT, untreated *N. gaditana*; NG T, treated *N. gaditana*; SFAs, saturated fatty acids, MUFAs; monounsaturated fatty acids; PUFAs, polyunsaturated fatty acids.

**Table 2 marinedrugs-20-00318-t002:** Chemical characterization of the ethanolic extracts from untreated and treated *N. gaditana*.

	NG UT	NG T
**Ethanolic extract yield**		
mg/mL extract	42.0 ± 0.99	79.0 ± 3.22
mg extract/g microalgae	177.7 ± 16.1	505.0 ± 30.2 ***
**Total polyphenols**		
µg GAE/mg extract	3.97 ± 0.93	21.6 ± 0.54 ***
µg GAE/g microalgae	780.9 ± 9.12	10,848.5 ± 502.7 ***
**DPPH**		
µg Trolox equivalent/mg extract	1.58 ± 0.30	2.89 ± 0.16 **
µg Trolox equivalent/g microalgae	254.6 ± 54.23	1632.4 ± 77.7 ***
**ABTS**		
µg GAE/mg extract	-	0.38 ± 0.01
µg GAE/g microalgae	-	193.68 ± 8.89
**Inhibition of lipid peroxidation**		
mUAA/mg extract	54.9 ± 7.98	89.9 ± 9.21 *
UAA/g microalgae	9.94 ± 2.35	45.0 ± 3.69 **

Results are represented as mean ± SEM. * *p* < 0.05; ** *p* < 0.01; *** *p* < 0.001. NG UT, untreated *N. gaditana*; NG T, treated *N. gaditana*; GAE, gallic acid equivalent; mUAA, milliunits of antioxidant activity.

**Table 3 marinedrugs-20-00318-t003:** In vitro digestibility of antioxidant and antiproliferative activity from untreated or treated *N. gaditana*.

	NG UT		NG T	
% Dry Matter Dialyzability	37.2 ± 1.84		60.1 ± 1.67 ***	
	**Dialyzate**	**Retentate**	**Dialyzate**	**Retentate**
Total polyphenols (µg GAE/mL)	68.0 ± 1.8	113.3 ± 2.6	245.1 ± 1.2 ***	385.1 ± 9.1 ***
Lipid peroxidation inhibition (UAA/g)	116.9 ± 34.2	112.6 ± 25.8	236.1 ± 17.4 *	98.1 ± 4.1
ABTS (µg GAE/mL)	3.77 ± 0.35	3.24 ± 0.24	4.76 ± 0.14 **	3.56 ± 0.06
Antiproliferative activity (µg/mL)	-	-	-	422.1

Results are represented as mean ± SEM. * *p* < 0.05, ** *p* < 0.01, *** *p* < 0.001. NG UT, untreated *N. gaditana*; NG T, treated *N. gaditana*; GAE, gallic acid equivalent; UAA, units of antioxidant activity.

**Table 4 marinedrugs-20-00318-t004:** Identification and potential digestibility of bioactive compounds from untreated or treated *N. gaditana*.

	RT	Name	[M-H]^-^	PPM	% Fit Conf.	Molecular Formula	Mass Fragment		
***N. gaditana* untreated**	1.9	Neohesperidose	325.1088	−14.5	70.55	C_12_H_21_O_10_	231.0415	217.0779	125.0277
	2.18	2,3,8, Trihydrosyanthraquinone	255.0313	7.8	99.27	C_14_H_7_O_5_	128.9957	120.0279	103.0257
	2.36	Abietin	341.1286	14.7	99.28	C_16_H_21_O_8_	309.141	231.0499	131.0376
	3.47	Epicatechin 3-O-(4-methylgallate)	455.0996	4	94.42	C_23_H_19_O_10_	332.0827	176.0189	119.9931
	4.66	Heteronemin	487.3072	2.5	99.49	C_29_H_43_O_6_	393.2438	293.1449	173.0310
	4.8	3 Methylflavone-8-carboxylic acid	279.0689	11.5	96.59	C_17_H_11_O_4_	248.0401	241.0796	227.0212
	8.09	Rehmanionoside C	387.2019	−9.3	51.99	C_19_H_31_O_8_	351.1888	325.1966	265.1518
	12.7	Oleandrin	575.3142	−13.6	68.05	C_32_H_47_O_9_	403.2211	301.2129	265.1341
**Dialyzate**	2.32	Caffeic acid 3-glucoside	341.0884	3.2	99.92	C_15_H_17_O_9_	272.0939	241.0191	221.084
	3.9	3-o-(α-L-oleandrosyl) oleandolide	529.2989	−4.5	93.22	C_27_H_45_O_10_	407.2008	315.2038	241.0772
	4.7	Heteronemin	487.3076	3.3	99.09	C_29_H_49_O_10_	393.2040	343.2649	315.2404
***N. gaditana* treated**	2.01	Quercetin-3-O-malonylglucoside	549.0905	4.6	95.83	C_24_H_21_O_15_	247.0692	227.0257	119.0652
	2.5	Asperulosidic acid	431.1163	7.4	51.51	C_18_H_23_O_12_	265.1312	248.0569	164.0708
	3.06	Petrosynone	455.2188	−7.5	63.9	C_30_H_31_O_4_	265.1512	125.0348	113.0126
	4.52	Oscillatoxin B1	589.3022	1.5	99.34	C_32_H_45_O_10_	393.2641	265.1356	227.0283
	5.22	Monensin B	655.4081	3.7	99.71	C_35_H_59_O_11_	593.304	590.3131	565.2737
	6.69	Sitoindoside IX	631.3138	3.2	95.96	C_34_H_47_O_11_	556.3114	393.2368	241.0997
	7.14	Withalongolide J.	635.3352	−12.4	66.64	C_34_H_51_O_11_	554.3320	407.1617	241.1268
	10.89	Physapubenolide	527.2603	−8	99.14	C_30_H_39_O_8_	405.1748	369.2156	293.2113
	11.13	Klymollin E	553.2725	13.7	98.62	C_28_H_41_O_11_	407.1414	369.2225	173.0797
	13.02	Forskoditerpenoside C	555.2905	18	95.53	C_28_H_43_O_11_	443.2647	265.1794	128.0935
**Dialyzate**	1.73	Quercetin-3-O-malonylglucoside	549.0878	−0.4	99.52	C_24_H_21_O_15_	279.0704	217.0761	207.0901
	3.09	Petrosynone	455.2212	−2.4	97.78	C_30_H_31_O_4_	265.1598	241.1031	157.0666
	8.44	Lasonolide G	893.5937	17.7	97.62	C_53_H_81_O_11_	652.3114	593.3041	448.3423

RT: retention time; PPM: error.

**Table 5 marinedrugs-20-00318-t005:** Formulation and nutrient composition of the experimental diets.

	CT	NG UT	NG T
** Diet formulation **			
Casein	133.3		
Methionine	5		
Sucrose	100		
Olive oil	70		
Mineral Mix, AIN-93G-MX	3.5		
Vitamin Mix, AIN-93-VX	10		
Choline bitartrate	2.5		
Cellulose	50	-	-
Untreated *N. gaditana*	-	200	-
Treated *N. gaditana*	-	-	200
Starch	625.7	475.7	475.7
** Nutrient composition **			
Kcal/kg diet	4010	3630	3610
Protein (g/kg)	118.7	195	193.1
Carbohydrates (g/kg)	734.2	535.9	528.8
Ash (g/kg)	25.9	56.3	71.1
Fat (g/kg)	66.9	78.5	80.6

CT, control diet; NG UT, untreated *N. gaditana*; NG T, treated *N. gaditana*.

**Table 6 marinedrugs-20-00318-t006:** Food intake, weight gain, fecal excretion, and digestive and metabolic utilization of protein.

	CT	UT NG	T NG
**Intake (g DM/d)**	18.2 (0.49) a	18.7 (0.49) a	18.9 (0.42) a
**Energy intake (Kj/d)**	77.2 (2.09) a	77.9 (3.20) a	79.0 (1.78) a
**Protein intake (g/d)**	2.16 (0.06) a	3.62 (0.15) b	3.68 (0.08) b
**Weight gain (g/d)**	5.18 (0.15) a	5.39 (0.45) a	4.63 (0.12) a
**PER**	2.40 (0.05) c	1.47 (0.07) b	1.27 (0.05) a
**FTI**	3.52 (0.07) a	3.58 (0.20) ab	4.10 (0.17) b
**Fecal weight (g DM/d)**	0.78 (0.10) a	3.00 (0.40) b	2.40 (0.20) b
**N intake (mg/d)**	345.7 (9.3) a	578.9 (23.8) b	588.1 (13.2) b
**Fecal N (mg/d)**	13.2 (1.10) a	192.5 (23.6) c	100.9 (7.4) b
**Urinary N (mg/d)**	113.0 (8.9) a	241.7 (21.5) b	240.7 (9.9) b
**Absorbed N (mg/d)**	332.5 (8.5) a	386.3 (10.6) b	487.2 (7.2) c
**ADC (%)**	96.2 (0.2) c	67.4 (2.9) a	83.0 (0.9) b
**Balance (mg/d)**	219.5 (6.6) b	144.6 (23.8) a	246.6 (7.6) b
**% R/A**	66.2 (2.2) c	37.1 (6.0) a	50.7 (1.7) b

Results are means and standard error of the mean (in parentheses) of 8 rats. CT, control group; NG UT, untreated *N. gaditana* group; NG T, treated *N. gaditana* group; PER, protein efficiency ratio; FTI, food transformation index; ADC, apparent digestibility coefficient; % R/A, retained to absorbed percentage. Means within the same line with different letters differ significantly (*p* < 0.05).

**Table 7 marinedrugs-20-00318-t007:** Fatty acid profile of plasma, erythrocyte membrane, and liver.

	Plasma			Erythrocyte Membrane			Liver		
	**CT**	**UT NG**	**T NG**	**CT**	**UT NG**	**T NG**	**CT**	**UT NG**	**T NG**
**Pamitic (C16)**	22.8 (0.7) a	24.0 (0.8) a	22.6 (0.5) a	33.2 (0.9) a	32.5 (0.6) a	33.4 (0.) 6 a	22.5 (0.6) a	21.0 (0.8) a	22.1 (0.4) a
**Palmitoleic (C16:1)**	2.14 (0.24) a	2.2 (0.3) a	1.84 (0.08) a	0.56 (0.09)	0.58 (0.15)	0.46 (0.13)	2.42 (0.29) a	1.93 (0.23) a	1.75 (0.23) a
**Stearic (C18:0)**	9.02 (0.50) a	10.0 (0.7) a	10.2 (0.3) a	14.8 (0.5) a	14.6 (0.47) a	15.1 (0.5) a	13.3 (0.8) a	18.2 (0.7) b	16.1 (0.6) b
**Oleic (C18:1n9)**	23.9 (1.3) a	21.9 (1.3) a	20.8 (1.6) a	12.7 (0.9) a	12.1 (0.37) a	11.3 (0.3) a	26.4 (1.5) b	19.2 (0.8) a	24.2 (1.6) b
**Octadecenoic (C18:1n7)**	5.95 (0.78) a	5.81 (0.44) a	5.83 (0.40) a	2.35 (0.55) a	2.91 (0.21) a	2.82 (0.38) a	4.97 (0.47) b	3.58 (0.28) a	3.80 (0.35) ab
**Linoleic (C18:2n6)**	11.2 (0.5) a	10.3 (0.6) a	11.2 (0.5) a	5.55 (0.25) a	5.12 (0.24) a	5.20 (0.33) a	8.61 (0.53) a	8.30 (0.40) a	8.04 (0.51) a
**Eicosatrienoic (C20:3n6)**	0.71 (0.23)	0.94 (0.19)	1.32 (0.16)	0.56 (0.13)	0.29 (0.07)	0.33 (0.11)	0.65 (0.08)	0.66 (0.02)	0.81 (0.05)
**Arachidonic (C20:4n6)**	17.2 (1.3) b	12.0 (1.5) a	18.4 (1.2) b	21.9 (0.5) a	20.3 (0.62) a	22.6 (0.5) a	14.9 (1.1) a	15.6 (0.7) a	15.3 (0.9) a
**Erucic (C22:1n9)**	-	-	-	-	-	-	0.20 (0.04) a	2.17 (0.73) b	0.27 (0.07) a
**Eicosapentaenoic (C20:5n3)**	0.37 (0.09) a	5.84 (0.73) b	1.04 (0.15) a	0.28 (0.10) a	2.86 (0.51) b	0.38 (0.14) a	-	-	-
**Nervonic (C24:1)**	-	-	-	2.53 (1.68) a	4.22 (0.17) b	2.92 (0.09) b	0.33 (0.08)	1.96 (0.12)	0.62 (0.05)
**Docosahexaenoic (C22:6n3)**	2.02 (0.19) a	2.71 (0.27) ab	3.05 (0.22) b	2.45 (0.34) a	2.48 (0.09) a	2.98 (0.26) a	3.28 (0.23) a	4.77 (0.33) b	4.61 (0.28) b
**Others**	4.48 (0.46) b	2.41 (0.35) a	3.89 (0.40) b	3.16 (0.32) b	1.99 (0.09) a	2.43 (0.23) ab	2.49 (0.15) a	2.69 (0.80) a	2.55 (0.16) a
**SFAs**	33.7 (0.9) a	35.3 (1.4) a	34.07 (0.7) a	49.9 (1.6) a	49.1 (0.6) a	50.6 (1.0) a	37.0 (0.4) a	40.4 (0.9) b	39.4 (0.6) b
**MUFAs**	32.2 (1.7) a	31.9 (1.5) a	29.02 (1.7) a	18.1 (1.5) a	19.8 (0.7) a	17.5 (0.5) a	34.4 (2.1) a	28.9 (0.7) a	30.7 (1.8) a
**PUFAs**	34.1 (1.3) a	32.8 (2.1) a	36.7 (1.3) a	31.9 (0.5) a	31.1 (0.7) a	31.9 (0.9) a	28.8 (1.7) a	30.8 (1.2) a	30.1 (1.4) a

Results are means and standard error of the mean (in parentheses) of 8 rats. CT, control group; UT NG, untreated *N. gaditana* group; T NG, treated *N. gaditana* group. Means within the same line with different letters differ significantly (*p* < 0.05).

**Table 8 marinedrugs-20-00318-t008:** Weight of different organs and tissues.

	CT	UT NG	T NG
Heart	0.34 (0.01) a	0.36 (0.04) a	0.32 (0.02) a
Cecum	0.18 (0.01) a	0.27 (0.01) b	0.25 (0.01) b
Colon	0.35 (0.02) a	0.34 (0.01) a	0.31 (0.04) a
Brain	0.63 (0.02) a	0.62 (0.01) a	0.61 (0.01) a
Epididymal fat	2.16 (0.17) a	1.94 (0.10) a	1.96 (0.18) a
Abdominal fat	2.37 (0.19) b	1.90 (0.13) ab	1.70 (0.16) a
Kidney	0.36 (0.01) a	0.41 (0.01) b	0.43 (0.01) b

Results are means and standard error of the mean (in parentheses) of 8 rats. CT, control group; UT NG, untreated *N. gaditana* group; T NG, treated *N. gaditana* group. Data are expressed as g/100 g of body weight. Means within the same line with different letters differ significantly (*p* < 0.05).

**Table 9 marinedrugs-20-00318-t009:** Hematic and biochemical parameters of rats fed untreated or treated *N. gaditana*.

	CT	UT NG	T NG
**WBCs** (×10^3^/µL)	3.11 (0.09) a	4.21 (0.48) a	3.71 (0.30) a
**RBCs** (×10^6^/µL)	6.99 (0.15) a	6.95 (0.12) a	6.95 (0.07) a
**HGB** (g/dL)	13.4 (0.25) a	14.4 (0.22) b	13.7 (0.16) ab
**HCT** (%)	34.5 (0.59) a	36.8 (0.65) b	35.7 (0.36) ab
**PLT** (×10^3^/µL)	602.6 (33.4) a	528.7 (22.9) a	618.8 (22.6) a
**MCV** (fL)	49.3 (0.77) a	53.0 (0.62) b	51.4 (0.40) b
**MCH** (pg)	19.2 (0.30) a	20.7 (0.28) a	19.8 (0.18) a
**MCHC** (g/dL)	38.8 (0.30) a	39.2 (0.22) a	38.4 (0.17) a
**RDW** (fL)	16.6 (0.24) a	16.2 (0.22) a	16.2 (0.26) a
**TGs** (mg/dL)	102.9 (16.7) a	116.8 (9.0) a	84.6 (11.0) a
**T-Chol** (mg/dL)	52.5 (4.0) a	63.5 (4.5) a	66.4 (2.9) a
**HDL-C** (mg/dL)	28.0 (3.6) a	39.3 (2.5) ab	45.8 (1.8) b
**LDL-C** (mg/dL)	13.3 (0.4) a	15.4 (1.0) a	17.4 (0.9) a
**Albumin** (g/dL)	3.13 (0.04) a	3.47 (0.03) a	3.47 (0.02) a
**Creatinine** (mg/dL)	0.28 (0.04) a	0.28 (0.03) a	0.28 (0.02) a
**AST** (U/L)	75.9 (14.8) a	93.8 (5.8) a	72.1 (4.5) a
**ALT** (U/L)	12.5 (1.6) a	22.3 (2.0) b	19.8 (1.6) ab
**γ-GT** (U/L)	1.90 (0.26) a	2.31 (0.43) a	2.65 (0.72) a
**ALP** (U/L)	153.6 (8.4) a	213.1 (12.5) b	169.0 (13.4) a
**P** (mg/dL)	9.41 (0.27) b	8.34 (0.17) a	7.98 (0.30) a

Results are means and standard error of the mean (in parentheses) of 8 rats. CT, control group; UT NG, untreated *N. gaditana* group; T NG, treated *N. gaditana* group; WBCs, white blood cells; RBCs, red blood cells; HGB, hemoglobin; HCT, hematocrit; PLT, platelet; MCV, mean corpuscular volume; MCH, mean corpuscular hemoglobin; MCHC, mean corpuscular hemoglobin content; RDW, red cell distribution width; TGs, triglycerides; T-CHOL, total cholesterol; HDL-C, HDL cholesterol; LDL-C, LDL cholesterol; AST, aspartate aminotransferase; ALT, alanine aminotransferase; γ-GT, gamma-glutamyl transferase; ALP, alkaline phosphatase; P, phosphorus. Means within the same line with different letters differ significantly (*p* < 0.05).

**Table 10 marinedrugs-20-00318-t010:** Urinary parameters of kidney functionality.

	CT	UT NG	T NG
pH	8.05 (0.20) b	6.10 (0.07) a	6.10 (0.07) a
Urinary weight (g)	3.64 (0.77) a	14.41 (1.93) c	6.94 (0.54) b
P (mg/dL)	3.47 (0.38) a	23.6 (8.0) b	54.7 (2.7) c
Uric acid (mg/dL)	23.0 (2.8) b	13.3 (1.0) a	9.5 (1.7) a
Albumin (g/dL)	0.097 (0.008) b	0.037 (0.004) a	0.056 (0.003) a
Creatinine (mg/dL)	66.8 (5.1) b	47.2 (3.6) a	72.8 (2.1) b

Results are means and standard error of the mean (in parentheses) of 8 rats. CT, control group; UT NG, untreated *N. gaditana* group; T NG, treated *N. gaditana* group; P, phosphorus. Means within the same line with different letters differ significantly (*p* < 0.05).

**Table 11 marinedrugs-20-00318-t011:** Effect of *N. gaditana* consumption on hepatic antioxidant activity.

Enzymes	CT	UT NG	T NG
Mn-SOD (U/mg protein)	19.0 (0.53) a	19.1 (0.25) a	21.6 (0.93) b
Cu/Zn-SOD (U/mg protein)	695.2 (12.8) a	761.5 (16.74) b	1062.0 (16.5) c
CAT (U/mg protein)	17.4 (0.8) a	20.6 (0.90) b	19.4 (0.70) b
GPX (nmol NADPH/min/mg protein)	9.66 (0.33) b	6.73 (0.41) a	12.2 (1.12) c

Results are means and standard error of the mean (in parentheses) of 8 rats. CT, control group; UT NG, untreated *N. gaditana* group; T NG, treated *N. gaditana* group; SOD, superoxide dismutase; CAT, catalase; GPX, glutathione peroxidase. Means within the same line with different letters differ significantly (*p* < 0.05).

## Data Availability

Not applicable.

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
