# Peer review of "In Vivo Nutritional Assessment of the Microalga Nannochloropsis gaditana and Evaluation of the Antioxidant and Antiproliferative Capacity of Its Functional Extracts"

_marinedrugs, 2022, doi:10.3390/md20050318_

Round 1

Reviewer 1 Report

The manuscript by Rosario Martínez and co-authors entitled «In vivo nutritional assessment of the microalga Nannochloropsis gaditana and evaluation of the antioxidant and antiproliferative capacity from its functional extracts» (Marine Drugs) provides the results of a large study on the effect of a treatment to increase the fragility of the cell wall on the bioavailability of its nutrients and functional compounds. The authors/researchers showed that functional extracts from treated N. gaditana exhibited higher antioxidant activity than the untreated control. Furthermore, the treated microalga induced hypo glycemic action, higher nitrogen digestibility, and increased hepatic antioxidant activity. In conclusion, N. gaditana has interesting potential as hepatoprotective, antioxidant, and anti-inflammatory, thus proving itself an ideal functional food candidate. Especially if the microalga is treated to increase the fragility of its cell wall before consumption.

My major concern is “a combined thermal and high-pressure treatment” that was used in this study to increase the fragility of algal cells. Since the patent pending, the techniques were not described; and such treatment caused a significant decrease in the lipid yield and, most importantly, polyunsaturated fatty acid contents, e.g., reduction in the DHA relative concentration from 42.6% to 1.26%. Usually, a very mild temperature hydrolysis under a deem light and an argon atmosphere is used for this purpose, so the details of the treatment used in this study should be described to understand a level of the potential oxidative stress (for lipids and fatty acids).

The authors did not specify what particular compound(s), extracted/left in the treated samples, were responsible for “the antioxidant and antiproliferative capacity”, although they provided some clear evidence for it. In my opinion, the manuscript should be significantly restructured/rewritten taking into consideration that nutritional quality of N. gaditana as a source of omega-3 PUGAs becomes very low (and cannot be considered as a supplement of these important compounds, that is commonly-used in aquaculture in the original form) after the treatment used (and the treatment should be described).

A minor comment: many mistakes in fatty acid names (e.g., Table 1 – “araquidonic” instead “arachidonic” and others along the text).

Author Response

Granada, May 3rd 2022

Dear Editor,

We appreciate the work of the referees who have enriched the manuscript enormously with their comments. Changes have been done according to their suggestions to achieve the desired improvements in manuscript quality. Furthermore, as the editor indicated us, changes have been made in two of the paragraphs in the materials and methods section to avoid problems with the similarities check. In addition, five new bibliographic citations have been included as well as new data from experiments carried out following the recommendations made by reviewer two.  

Kind regards,

Jesus M. Porres, PhD

Department of Physiology

Universidad de Granada

Spain

Regarding the editor comments, we enclose the answers to the questions raised:

REVIEWER 1.

- My major concern is “a combined thermal and high-pressure treatment” that was used in this study to increase the fragility of algal cells. Since the patent pending, the techniques were not described; and such treatment caused a significant decrease in the lipid yield and, most importantly, polyunsaturated fatty acid contents, e.g., reduction in the DHA relative concentration from 42.6% to 1.26%. Usually, a very mild temperature hydrolysis under a deem light and an argon atmosphere is used for this purpose, so the details of the treatment used in this study should be described to understand a level of the potential oxidative stress (for lipids and fatty acids).

Answer: Following the reviewer´s recommendation, we have extended the materials and methods section in subsection 4.1. to include more information regarding the pressure and temperature conditions used for the microalgal treatment. Page 16, Ln. 528-534. “Nannochloroposis gaditana was kindly provided by Endesa Generación, Carboneras, Almería, Spain). A combined thermal and high-pressure treatment, via steam explosion at temperatures between 160-190°C in time intervals between 5 and 30 minutes, was implemented by the Instituto de la Grasa (Spanish National Research Council, Sevilla, Spain) to increase the fragility of N. gaditana (patent pending) and make it more susceptible to disruption during extraction processes and digestion by the experimental animals”.    

- The authors did not specify what particular compound(s), extracted/left in the treated samples, were responsible for “the antioxidant and antiproliferative capacity”, although they provided some clear evidence for it. In my opinion, the manuscript should be significantly restructured/rewritten taking into consideration that nutritional quality of N. gaditana as a source of omega-3 PUGAs becomes very low (and cannot be considered as a supplement of these important compounds, that is commonly-used in aquaculture in the original form) after the treatment used (and the treatment should be described).

Answer: The specific compounds identified in the microalgae by UPLC-MS and the use of the Chemspider database are described in Table 4 and section 2.4 of the Results section. The technological treatment used led to the detection of new compounds with respect to the untreated algae that are described in detail. We have specified in the results and discussion sections the compounds that may be related to the antiproliferative and antioxidant action, also including in the discussion section new bibliographic citations that support our results. Page 14, Ln. 404-408. “These bioactive compounds may be related to at least three beneficial biological properties of the microalga such as its antioxidant, antidiabetic, and antiproliferative action. In fact, the antiproliferative action of heteroneminin and the antioxidant activities of caffeic acid and quercetin-3-O-malonylglucoside have been reported [22-24]”.

Answer: Regarding the comment on the nutritional utilization of the microalgae in terms of its omega-3 content, the authors wish to emphasize that in this study it was important firstly to highlight the nutritional potential of the microalgae N. gaditana, and, secondly, to provide data on a technological treatment that allows improving the bioavailability of some of the nutrients and bioactive compounds that have not been studied until now. In this sense, we have reported that the untreated microalga is a good source of protein, minerals and n-3 fatty acids. Specifically, the latter are bioaccessible and incorporated into the plasma, the erythrocyte membrane and the liver, from where their beneficial anti-inflammatory action and in the treatment of cardiovascular disease can be derived. To this beneficial effect, those described for protein and minerals can be added, which make this microalga a promising source of nutrients and functional compounds. Furthermore, the treatment has led to improvements in protein bioavailability and bioaccessibility of components with antioxidant effect, which is a clear additional benefit on its nutritional and functional value with respect to the untreated microalga. All this despite the fact that, as the reviewer indicates, the treatment makes N. gaditana a poorer source of n-3 PUFAs. The positive effects described in relation to protein and bioactive compounds have been explained in detail in the discussion section and we consider that they adequately justify the implementation of the selected technological treatment. However, following the reviewer's recommendations, we have restructured the discussion and included a paragraph to clarify that it is the untreated microalgae (or the microalgae subjected to a milder treatment) that should be used if a supplement of n-3 fatty acids is sought. Page 15, Ln. 473-480. “This fatty acid has been credited for interesting metabolic benefits of its derivatives in the inflammatory process, cardiovascular disease or diabetes [38]. In contrast the technological treatment caused a reduction in fat content and changes in the fatty acid profile to-wards a more saturated pattern in the treated microalgae, leading to significantly lower levels of EPA in the tissues studied compared to the untreated algae. Therefore, if the beneficial effects of N. gaditana n-3 fatty acids are sought, the untreated microalgae (or, alternatively, treated using a milder technological process) must be used.”.

- A minor comment: many mistakes in fatty acid names (e.g., Table 1 – “araquidonic” instead “arachidonic” and others along the text).

Answer: a carefully spell check have been done to correct mistakes in fatty acid names pointed out by the reviewer.

Reviewer 2 Report

The interesting data of the research study show a new approach to the potential benefits of  the N. gaditana  technologically treated microalgae as  nutritional supplement or in the prevention and treatment of disease. Indeed the changes in fatty acid profile towards a more saturated pattern cause the improvement in the antioxidant properties and reduced cytotoxicity in vitro and in animals.

However it is not clear in introduction and discussion what the disease refers to the authors. NAFLD, the nonalcholic fatty liver disease or other liver diseases? Please, specify better alongside the manuscript.

Moreover, the authors should better underline:

1) what is the cytotoxicity of the new compound on human non cancer (colon or other) cells?

2) what will be the impact of these results on human diet and the feasibility of commercialization, if there will.

Author Response

Granada, May 3rd 2022

Dear Editor,

We appreciate the work of the referees who have enriched the manuscript enormously with their comments. Changes have been done according to their suggestions to achieve the desired improvements in manuscript quality. Furthermore, as the editor indicated us, changes have been made in two of the paragraphs in the materials and methods section to avoid problems with the similarities check. In addition, five new bibliographic citations have been included as well as new data from experiments carried out following the recommendations made by reviewer two.  

Kind regards,

Jesus M. Porres, PhD

Department of Physiology

Universidad de Granada

Spain

Regarding the editor comments, we enclose the answers to the questions raised:

REVIEWER 2.

The interesting data of the research study show a new approach to the potential benefits of  the N. gaditana  technologically treated microalgae as  nutritional supplement or in the prevention and treatment of disease. Indeed the changes in fatty acid profile towards a more saturated pattern cause the improvement in the antioxidant properties and reduced cytotoxicity in vitro and in animals.

- However, it is not clear in introduction and discussion what the disease refers to the authors. NAFLD, the nonalcholic fatty liver disease or other liver diseases? Please, specify better alongside the manuscript.

Answer: Thank you for your comments. According to your suggestions we have specified in the introduction and extended in the discussion the diseases in which dietary treatment with microalgae could be of help. Mainly the metabolic syndrome, cancer and non-alcoholic fatty liver disease.

Introduction: Pages 1-2, Ln. 46-50. “Among them, those exerting benefits on obesity, cancer, and metabolic disorders like the metabolic syndrome, major epidemics in current society, have attracted special attention. Furthermore, hepatic functionality in alterations like the non-alcoholic fatty liver disease can be improved by plant-derived foods or functional extracts [1]. Such increased interest is reflected in a growing number of new patents for the food industry [2].”

Discussion: Page 13, Ln. 370-372. “The results of the present study represent a new approach to the potential benefits of N. gaditana consumption either as a nutritional supplement or in the prevention and treatment of diseases like cancer, metabolic syndrome or non-alcoholic fatty liver disease”.

Conclusions: Page 21, Ln. 769-770. “thus making it an ideal candidate functional food for preventing and/or managing alterations like metabolic syndrome and associated non-alcoholic fatty liver disease”  

Moreover, the authors should better underline:

1) what is the cytotoxicity of the new compound on human non cancer (colon or other) cells?

Answer: New experiments have been performed following the reviewer´s suggestion using CCD18 human non-tumor colon, and MCF-10A human non-tumor breast epithelial cells. The results show a significantly lower antiproliferative effect of the treated N. gaditana extract in the three cell lines (T84 human colon adenocarcinoma, CCD18 human non-tumor colon, and MCF-10A human non-tumor breast epithelial cells) tested when compared to the untreated microalga extract. The cytotoxic action of untreated N. gaditana extract did not vary greatly among the three cell lines tested. Results are now included in the results section. Page 4, Ln. 139-146. “The ethanolic extracts from untreated or treated N. gaditana were assayed in a T84 colon cancer cell line, as well as in CCD18 colon or MCF-10A mammary non-tumor cell lines, to assess its antiproliferative activity. As shown in Figure 1, the antiproliferative activity of treated N. gaditana extract was significantly reduced compared to the untreated microalga as seen by the significantly higher IC50 achieved by the former in the tumoral cell line T-84 and the absence of IC50 in the CCD18 and MCF-10A non-tumor cell lines. Regarding untreated N. gaditana extract, a lower IC50 value was obtained in CCD18 vs T84 colon cell line whereas the highest IC50 was obtained for the MCF-10A cell line”. 

2) what will be the impact of these results on human diet and the feasibility of commercialization, if there will.

Answer: Following the reviewer´s recommendation, we have added a new paragraph to describe the impact of our results on human nutrition and the feasibility of commercializing the product. Page 16, Ln. 519-525. “The results of this study can have a significant impact for human nutrition, since novel foods are continuously been tested to increase the quality of current dietary patterns. This can be done providing essential nutrients to improve the nutritional status of the population, or through the supplementation of bioactive compounds with beneficial health-related effects, both aspects covered by N. gaditana. The microalga is already commercialized, although not widely distributed, and our results can give this food product added value and new potential uses”.

Round 2

Reviewer 1 Report

The manuscript has been amended according to the comments